# Convolution Can Incur Foveation Effects

Jun Yuan, Bilal Alsallakh, Narine Kokhlikyan, Vivek Miglani, Orion Rebliz-Richardson

**Abstract.** We demonstrate how boundary treatment in convolutional networks can incur foveation effects: Impacted pixels have fewer ways to contribute to the computation than central pixels. Different padding mechanisms can either eliminate or aggravate these effects (Alsallakh et al. 2021). This is made obvious via a web-based interactive visualization, available at https://mind-the-pad.github.io/

**Benefits Of the Proposed Format**

In the aforementioned work (Alsallakh et al. 2021), foveation effects were illustrated for a few boundary treatment methods via a lengthy 18-page appendix.

- **Presentation:** The visualization is far better at communicating foveation effects:
    - It better reveals how the padding mechanism shapes these effects, e.g. 0-padding, reflection, circular, or partial convolution (Liu et al. 2018).
    - It better reveals, under which padding method these effects are insensitive to dilation and kernel size.
    - It enables examining and understanding the effect for every pixel the user clicks, whereas the PDF appendix only focuses on exemplar pixels.
    - It is suited for asynchronous research presentation, for virtual conferences, and for education.
- **Reviewing:** The animated visualization is easier to review:
    - It is significantly more concise.
    - It provides visual intuition that helps convince the reviewers.

**Interoperability**

The interactive visualization can be converted to animated GIFs to showcase particular examples. It does not fit in PDF format, however, snapshots can be embedded in PDF to illustrate particular cases. can be converted to animated GIFs to showcase particular examples.

**Potential Limitations**

- While version control is possible in the GitHub repository, the effect of changes is not always easy to realize. In particular, it is not straightforward to compare two versions, as in the case of PDF.

- Durability: Changes in web standards might impact the layout of our animated visualizations. While we strictly used standard HTML elements, we cannot guarantee the layout will be correctly rendered under potentially significant changes over decades.

**Related Work**

Our work is inspired by the following interactive visulizations of CNNs:

- The Distill article on Computing Receptive Fields of CNNs (Arajo et al, 2016). It demonstrates how the kernel size, the stride, and the padding used in each layer impact the receptive field of CNNs.
- The Convolution Visualizer (Yang, 2018) which illustrates how the output pixels of conv layer are computed, based on the input pixels and the hyperparameters of the convolutilnal layer.
- The Distill article on Deconvolution and Checkerboard Artifacts (Odena et al 2016). It demonstrates how certian filter size and stride choices can lead to checkerboard artifacts in the CNN's innards and output.
- The CNN Explainer (Wang et al, 2020), which demonstrates the power of animated visualization in communicating the innards of CNNs.

None of the above-mentioned projects demonstrates how different padding algorithms work and how they influence the foveation behavior of CNNs. Furthermore, previous analysis of CNNs with foveated input (Deza and Konkle, 2020) does not take padding into consideration.

**Accessibility Statement**

Our animated visualizations rely on a simplified point-and-click user interface. At its current accessibility level, our implementation expects users to operate such a graphical interface. Steps taken to improve accessibility concerns:

- We reduced user affordance to increase accessibility to users with motor disabilities. Our UI does not demand users to enter any text or to drag sliders. It is composed solely of buttons that enable the user to switch between padding mechanisms and pre-defined cases (e.g. dilation = 2, kernel size = 5x5).
- We used color scales that make different values distinguishable to color-blind users (see colorbrewer2.org).
- The UI works on tablets and smartphones, besides desktop and laptop computers.

We will enable users to open issues and submit pull requests on GitHub to report issues with operating the UI, including exceptions to intended accessibility levels, aiming to continuously improve accessibility.

## Practical Singifcance and Implications

Our work serves as a guide to practitioners and researchers on the implications of different padding algorithms on how the boundary will be treated in CNNs. This aims to increase awareness about potential foveation effects and to make informed decisions that fit the task.

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
