# OpenReview forum: "Convolution Can Incur Foveation Effects"
_ICLR.cc/2021/Workshop/Rethinking_ML_Papers/Exhibit_and_Workflow — Rethinking ML Papers - ICLR 2021 workshop Poster_

### Official Review · Reviewer_vLPv · 2021-03-27
**A Succinct Demonstration of Padding and Subsequent Foveation in Convolution**

**Accessibility:**

Score of 5 (Exceptional): Submission identifies and articulates accessibility matters, provides justifications for the proposed paradigm, and declares the limitations.

**Groundsforrejection:**

There's not really much of a lit review involved in the demo, but I would argue that it doesn't seem terribly important to this artifact either. Many of the padding approaches are so straightforward that attribution seems dubious. It does link out to one approach that is fairly non-obvious.

**Litreview:**

Score of 3 (Neutral): The submission acknowledges previous work, but does not necessarily explain how the submission differentiates itself (i.e we want to avoid the “deluge of citation” strategy, leaving the reviewer to click through references and figure this part out for themselves).

**Problemstatement:**

Score of 2 (Needs Improvement): The submission clearly has potential or credibility, but still fails to state the problem addressed clearly.

**Relevance:**

Score of 4 (Strong): The submission directly addresses a theme of the workshop, and does so in a very professional manner.

**Results:**

Score of 4 (Strong): Submission is very well structured and follows all the criteria (i.e. clarity, novelty, interactivity, and coherency). However, practical significance/theoretical implications are not discussed.

**Reviewerconfidence:**

5 - I'd like to see a little more of a problem statement added.

**Reviewtext:**

The interactive demo provides a clear demonstration of the amount and behavior of different foveation effects incurred by choosing padding techniques for convolution. This gives a nice explanation of the potential effects that occur with different choices of padding algorithms, with only modest interaction needed.  This work serves a straightforward example of non-paper artifacts to help with explainability and pedagogy. The demo is clearly aimed at expert audiences as it does not provide very much of a problem statement related to foveation, instead referring readers to the associated ICLR paper.  A small problem statement discussing downstream issues of foveation would greatly improve the overall value of this demo!

Minor comments:
A small label "Common Padding Approaches" or "Common Padding Techniques" above the set of explorable padding techniques would make this clearer to non-experts.

It may be worth numbering the "Input (left top corner of an image)" and "Step-by-step illustration". I was immediately drawn to the play/pause controls rather than first selecting a pixel (even despite the helpful hand glyph and instructions).

**Score:**

Accept: The reviewer believes the submission provides a novel and reliable scheme to improve science communication but needs improvement.

---

### Official Review · Reviewer_xzfP · 2021-03-30
**Effective visualisation, clarity may be improved**

**Accessibility:**

Score of 5 (Exceptional): Submission identifies and articulates accessibility matters, provides justifications for the proposed paradigm, and declares the limitations.

**Litreview:**

Score of 3 (Neutral): The submission acknowledges previous work, but does not necessarily explain how the submission differentiates itself (i.e we want to avoid the “deluge of citation” strategy, leaving the reviewer to click through references and figure this part out for themselves).

**Problemstatement:**

Score of 4 (Strong): The submission sets a very strong example of how to address the problem, which should be relevant to the workshop themes.

**Relevance:**

Score of 4 (Strong): The submission directly addresses a theme of the workshop, and does so in a very professional manner.

**Results:**

Score of 3 (Neutral): Submission is well designed and provides a good level of coherency/novelty/interactivity.

**Reviewerconfidence:**

I feel I am around 4, on a scale of confidence 1-5. I have carefully review the submission and the visualisation tool provided, and I have as well partly read the paper linked to the visualisation. Furthermore, the topics of the submission are related to my own fields of research. This contributes positively to my confidence. However, I have not carefully read the associated paper (an ICLR submission), reason why I am not 100 % confidence about my evaluation.

**Reviewtext:**

[Due to the character limit, I had to significantly cut down and simplify my feedback. I apologise to the authors for the concise wording and lack of details]

This submission presents a website with an interactive visualisation of the effect of padding on convolutional layers. The tool supplements a paper, by the same authors, to be presented at ICLR 21. The paper analyses the impact of different padding methods on the relative contribution of each pixel on the computation of conv. layers.

Strengths:

+ The visualisation tool is effective at illustrating and reducing the cognitive load for understanding the main message of the paper.

The main weaknesses relate to the clarity of the website:

- The primary purpose of the website and visualisation tool is not immediately clear. Consider, for example, setting a subtitle: "The number of convolutional operations different pixels are involved in", or "Are all pixels involved in the same number of convolutional operations?".
- Consider adding a title, for instance "Padding methods", on top on the list of padding methods.
- What does the image associated with each padding method represent? I understand it now, but not at first glance. There is no title or caption describing that figure and the colourbar is too small to be readable.
- Using these images to identify the padding methods may be confusing: consider identifying them with what they do (the prior knowledge of users about padding), instead of the count of operations (what users are to learn from the visualisation tool and paper).
- Why having separate panels for the "Input" and "Step-by-step illustration"? Consider merging these two into one single panel, so that we can directly see the effect of padding, click on the pixel and see the animation on the same panel.
- Why do the cells of "Input (left top corner of an image)" turn white when hovering with the cursor over them?

Despite these comments about clarity, I positively evaluate the submission overall.

**Score:**

Accept: The reviewer believes the submission provides a novel and reliable scheme to improve science communication but needs improvement.

---

### Meta-Review · Area_Chair_UqPb · 2021-03-31

**Recommendation:** Accept
**Confidence:** 4

**Metareview:**

The work demonstrates "how boundary treatment in convolutional networks can incur foveation effects:  Impacted pixels have fewer ways to contribute to the computation than center pixels." A nice submission that raises awareness about foveations and different padding choices. However,  the workshop submission needs improvement in several aspects. I strongly recommend addressing the reviewer's concerns around the following:

- A major concern is that while the submission clearly has potential or credibility, but still fails to state the problem addressed clearly.
- Clarity of the website is lacking, please see reviewers' detailed comments on what needs to be addressed.
- The submission acknowledges previous work, but does not necessarily explain how the submission differentiates itself - Comparing and contrasting important prior work in some detail might help alleviate these concerns.
- It would be important to make this submission self-contained as much as possible. At present "the demo is clearly aimed at expert audiences as it does not provide very much of a problem statement related to foveation, instead referring readers to the associated ICLR paper." Reading the reviews on the ICLR paper linked, it seems the original work is well done and more complete. I highly recommend presenting this work here in a short but complete (and self-contained) manner.
- See more details in reviewer comments.

---

### Decision · Program_Chairs · 2021-04-01

Accept (Poster)